# How Unawareness of Weight Excess Can Increase Cardiovascular Risk?

**DOI:** 10.3390/jcm11174944

**Published:** 2022-08-23

**Authors:** Magdalena Zalewska, Jacek Jamiołkowski, Małgorzata Chlabicz, Magda Łapińska, Marlena Dubatówka, Marcin Kondraciuk, Adam Hermanowicz, Karol Adam Kamiński

**Affiliations:** 1Department of Population Medicine and Lifestyle Diseases Prevention, Medical University of Bialystok, 15-259 Bialystok, Poland; 2Department of Invasive Cardiology, Medical University of Bialystok, 15-259 Bialystok, Poland; 3Department of Pediatric Surgery and Urology, Medical University of Bialystok, 15-274 Bialystok, Poland; 4Department of Cardiology Medical, University of Bialystok, 15-259 Bialystok, Poland

**Keywords:** obesity, diet, cardiovascular risk

## Abstract

Background: Obesity is a chronic disease with high prevalence in all age groups. Many overweight and obese people seem to be unaware of excess body weight. Aim: Analysis of people affected by the misperception of excess body weight and their eating behaviors simultaneously with selected health parameters. Methods: The study was conducted in 2017–2019 among 658 participants aged 20–79 from the population study—Bialystok PLUS (Poland). Results were based on clinical examinations and questionnaires. Results: Unawareness of overweight and obesity is common among adults (21.7%). Participants unaware of their overweight and obesity presented much higher risk factors. A high cardiovascular risk profile was observed more often among people not aware of overweight and obesity than among normal weight people (23.0% vs. 10.0%) as well as more common asymptomatic carotid artery atherosclerosis (49.7% vs. 31.3%). The subjective perception of overweight and obesity based on BMI (body mass index) was equal to 26.4 kg/m^2^ in women and 27.9 kg/m^2^ in men. The assessment of their diet was less favorable than that of people with normal weight. Conclusions: Unawareness of one’s excessive weight and its health consequences may lead to hesitancy to apply a healthy lifestyle and hence increase the cardiovascular risk in a substantial part of society. Therefore, it should be considered a part of the cardiovascular disease risk spectrum. Measurement of BMI and discussion about its health implications should be a routine procedure during healthcare contacts.

## 1. Introduction

Globally, obesity is recognized as a significant public health problem. It is most accurately defined as the abnormal or excessive accumulation of adiposity to the extent that health may be impaired [1]. Worldwide, obesity has nearly tripled since 1975. In 2016, 39% of adults were overweight, and 13% were obese [2]. In Poland, the prevalence of overweight and obesity has dramatically increased in recent years. In 2016, 53% of women and 68% of men in Poland were overweight, and 23% of women and 25% of men were obese [3].

Obesity is the cause of numerous complications, including cardiovascular diseases, type 2 diabetes, osteoarthritis, liver and respiratory disease, and cancers [4,5,6]. Long-term obesity increases the risk of dyslipidemia and systemic inflammation, which could be common pathways to complications, arguably the most important of which are diabetes and vascular disease [7]. There is a link between excess abdominal fat and the risk of cardiovascular disease (CVD) and other cardiometabolic diseases [8,9,10]. Obesity and especially abdominal type of fat distribution are also associated with left ventricular dysfunction, quality of life, and a reduction in life expectancy [11,12,13,14]. The risk of obesity disease is significantly influenced by lifestyle, including diet, consisting primarily of too high consumption of animal fat, simple sugars, salt, and too low consumption of vegetables, fruits, and whole-grain cereal products [15,16]. Research shows that body weight is often perceived inadequately, understated, or overestimated [17,18]. The perception of nutritional status can be determined by many factors, such as age, sex, education, social status, social norms, personality traits, or one’s own life experiences in the family or the environment [19,20]. It has been shown that failure to identify as overweight or obese is more common among those with more overweight people in their environment.

An important issue is the lack of awareness of excess body weight and the resulting problems. There are overweight and obese people who accept their own body weight, considering it correct. Such misclassification of body weight may be related to improper health behaviors or a lack of motivation to change the lifestyle. This growing problem is related to, among other things, difficulties in its prevention and treatment.

The aim of the study was to analyze people affected by the misperception of excess body weight and their eating behaviors simultaneously with selected health parameters.

## 2. Materials and Methods

The Bialystok PLUS study was conducted in 2017–2020 on a sample of Bialystok residents aged 20–79 years old [21]. According to the 2017 Central Statistical Office data, the number of residents in Bialystok was 274,746. There were 206,784 in the age group 20–79. Randomly selected residents (1497) were invited to participate in the study. Of the invited, 6 people died before joining the study. In total, 678 participants were examined (Figure 1).

Anthropometric measurements were made, including height, weight, waist circumference, and hips. Blood pressure (BP) was measured while participants sat after a minimum rest period of 5 min using the oscillometric method and an MG Comfort instrument from Omron Healthcare Co., Ltd. MG Comfort (HEALTHCARE Co., Ltd. Terado-cho, Muko, Kyoto, Japan). Body mass index (BMI) was calculated as weight in kilograms divided by height in meters squared. Waist-to-hip ratio (WHR) was calculated as a ratio between waist and hips circumference. Body adiposity index (BAI) comprised both hip circumference and height according to the following equation:$$\mathrm{BAI}=\frac{hip\;circumference\;\left[\mathrm{cm}\right]}{heigth\left[m\right]\times \sqrt{height\left[m\right]}}-18$$

Body composition measurements were performed by GE Healthcare Lunar Dual Energy X-ray Absorptiometry (DEXA), with total body weight divided into 3 compartments: bone, fat mass, and lean mass. Total (TF), gynoid (G), and android (A) fat were measured automatically. The homeostasis model assessment of insulin resistance (HOMA-IR) was calculated according to the equation: fasting insulin µU/mL) × fasting glucose (mmol/L)/22.5. Diabetes was diagnosed in people with a glucose level at 120 min in OGTT ≥ 200 mg/dL.

The Systematic Coronary Risk Estimation (SCORE) system was recalibrated in Poland. Thus, we used the Pol-SCORE system to assess the 10-year risk of fatal CV disease based on the following risk factors: age, gender, smoking, BP, and total cholesterol.

The study included a questionnaire interview considering socio-economic factors, family history, obesity history, lifestyle interview, and dietary assessment.

The awareness of overweight and obesity was defined based on their knowledge about their own overweight and obesity. We compared the BMI value obtained based on anthropometric measurements: body weight and height measurement with the participants’ answers to the question: “*Do you think you are overweight or obese?*”. Based on the assessment of body weight perception and BMI, the following groups were distinguished:normal weight—normal weight in self-assessment and normal weight according to BMI (BMI < 25);unawareness of overweight and obesity—normal weight in self-assessment, but overweight/obesity according to BMI (BMI ≥ 25);awareness of overweight and obesity—overweight/obesity in self-esteem and normal weight according to BMI (BMI < 25);ignorance of the correct body weight—overweight/obesity in self-assessment and overweight/obesity according to BMI (BMI ≥ 25).

Participants with normal weight who declared self-perceived overweight were excluded from further analysis due to the small size of this group (N = 20). Those people who reported that their body weight was normal but anthropometric measures of overweight and obesity show differently we defined as “unaware of overweight or obesity”, and those who correctly identified their nutritional status were categorized as “aware”.

Based on the food frequency questionnaire (FFQ), we created a score of appropriate diet. The participants reported the frequency of consumption of food based on 6 levels of frequencies: 1—a given product was not eaten at all, 2—the product was consumed less than once per month, 3—the product was consumed 1–2 times per month, 4—the was product consumed once per week, 5—the product was eaten 2–3 times per week, 6—the product was eaten 4–6 times per week, 7—the product was eaten 7 times per week and more often. Based on The National Institute of Public Health–National Food and Nutrition Institute recommendations [22], 1 or 2 points were awarded for consuming the products that were recommended (see Table 1). Negative points −1 or −2 points were awarded for the consumption of non-recommended products or those that should be restricted in the healthy diet. Points calculated from all food categories are added and compose a score of a recommended diet. The analysis of the diet of the study participants based on the FFQ method takes into account the selected products most often eaten broken down into recommended and non-recommended according to the principles of proper nutrition.

### 2.1. Ethical Issues

Ethical approval for this study was provided by the Ethics Committee of the Medical University of Bialystok (Poland) on 31 March 2016 (approval number: R-I-002/108/2016). The study was conducted in accordance with the Declaration of Helsinki, and all participants gave written informed consent.

### 2.2. Statistical Analysis

Descriptive statistics for quantitative variables are presented as means and standard deviations and as counts and frequencies for qualitative variables. Comparisons of continuous variables between more than two subgroups were performed using the Kruskall–Wallis test. Post hoc pair comparisons were made using the Dwass–Steel–Critchlow–Fligner test [23]. Comparisons of categorical variables between subgroups were performed using Pearson’s chi-square test with post hoc tests for two proportions with Bonferroni correction for multiple comparisons. The perception of overweight and obesity in the study group was shown using ROC analysis. Areas under ROC curves between different overweight indicators were compared using nonparametric approach [24,25]. Optimal cut-off values were determined by maximizing value of Youden’s coefficient.

Statistical hypotheses were verified at a 0.05 significance level. The IBM SPSS Statistics 26.0 statistical software (Armonk, NY, USA) was used for all calculations.

## 3. Results

### 3.1. Socio-Demographic and Clinical Characteristics

The baseline characteristics of the study population are shown in Table 2.

In the study, the mean age of participants was 48.3 ± 15.2 years. Women prevailed in a group of participants with a normal weight (66.4%), while men predominated in the unawareness of overweight and obesity (64.3%). Men with the awareness of overweight and obesity constituted 49.4% of the study population. Differences in terms of gender participation in the study groups were statistically significant (*p* < 0.001). Most participants with normal weight were <40 years of age. Patients who were not aware of their overweight and obesity were significantly older than people with normal weight. The differences in age groups in the study groups were statistically significant (*p* < 0.001). The low level of education (elementary and vocational based on elementary school) was declared mostly by respondents who were not aware of their obesity; in contrast, higher education (bachelor’s degree and university) dominated in the group with normal weight.

Maternal obesity declared by 13.7% and father’s obesity by 10.5% of all participants. Childhood obesity reported 17.1% of participants. Over one-third of individuals declared weight gain over the last year. Family history, father’s, sister’s, and brother’s obesity or obesity in childhood were most often confirmed by people aware of overweight and obesity (see Appendix A). Nearly half of the people in this group declared unwanted weight gain over the last year. In the group with normal weight, this percentage was 24.9, and in the group with unaware overweight and obesity, this percentage was 28.0. This was not objective weight gain, but again self-perceived one.

The compared groups differed significantly in terms of BMI and WHR. As shown in Table 3, there was a significant difference in mean percent fat, mean tissue thickness, total fat mass, android fat mass, total lean mass, android fat mass, gynoid fat mass by DEXA between groups. Individuals with unawareness of overweight and obesity had the highest total bone mass. They also had significantly higher fat mass and percentage than people with normal weight. In this group of people, the average percentage of lean mass was significantly lower than in people with normal weight and significantly higher compared to people’s awareness of overweight and obesity.

In Table 4, we present clinical and biochemical characteristics compared between individuals with normal weight, unawareness of overweight and obesity, and awareness of overweight and obesity. Patients unaware of their overweight and obesity presented significantly worse blood lipids, insulin resistance and blood pressure parameters than participants with normal weight.

We confirmed diabetes mellitus from the interview among 7.4% of participants, more often among people unaware of overweight and obesity than with normal weight (7.1% vs. 2.0%). The presence of atherosclerotic plaques in the carotid ultrasound was confirmed in 31.3% of people with normal weight, 49.7% with unawareness of overweight and obesity, and 56.4% with awareness of overweight and obesity (*p* < 0.001), (see Appendix A).

The differences in high-density lipoprotein (HDL) cholesterol and low-density (LDL) cholesterol between the three groups compared were statistically significant (*p* < 0.001). People with unawareness of overweight and obesity had high-density lipoprotein (HDL) cholesterol lower and low-density (LDL) cholesterol that were higher compared to normal weight people (Figure 2 and Figure 3).

The assessment of cardiovascular risk in the studied groups differed significantly (*p* < 0.001). Low risk most often was observed in participants with normal weight. People unaware of overweight and obesity more often had higher cardiovascular risk than patients with normal weight. Importantly, high cardiovascular risk profile was observed more often in the group not aware overweight and obesity than in group with normal weight (Figure 4).

### 3.2. Overweight and Obesity Perception and Diet Assessment

We decided to estimate what are the cut-off points in terms of subjective perception of overweight and obesity in the study population was shown using ROC analysis (see Appendix A). As shown in Table 5, the actual subjective perception point of overweight and obesity based on BMI in women was 26.4. The WHR rate of obesity in the perception of the women was 0.8, and the WC was 81.8 cm. The percentage of body fat indicating overweight and obesity in women was 38%.

Among men, the individual perception point of overweight and obesity based on BMI was 27.9. The WHR rate of obesity in men in their perception was 1.0, and the WC was 90.3 cm. The percentage of body fat that indicates obesity in men was 29%. In both sexes, BMI, WC and average percentage of fat tissue better described perception of overweight and obesity than WHR measurement. The WHR index is a significantly weaker indicator in the assessment of obesity than the others (BMI, WC, average percentage of fat) (Table 5). In addition, the subjective perception of overweight and obesity was, to the greatest extent, based on the BMI index, both among women (AUC = 0.9153) and men (AUC = 0.9125).

The participants’ lifestyles with normal weight, unawareness of overweight, and obesity with awareness overweight and obese are presented in Table 6. There were no statistically significant differences between the groups regarding current cigarette smoking, sleep time, and sleep quality self-assessment. In turn, self-assessment of the subjects’ diet showed statistically significant differences between the groups. Participants with unaware overweight and obesity rated their diet better compared to others (*p* = 0.001). That group participated in the spirometry tests more often than the others (*p* = 0.029). Although overweight and obese people were more likely to report exertional dyspnea with daily activities than those unaware of being overweight and obese (5.6% vs. 17.3%). People with normal weight less frequently than the others participated in preventive examinations of blood cholesterol assessment.

Participants unaware of their overweight and obesity described their diet as better than participants aware of their excessive body mass. In order to evaluate their diet more objectively, we calculated the score of recommended diet for each participant as described in methods section. People with normal body weight had the most points, which means their diet was the most appropriate. Diet score in participants unaware of their overweight and obesity was significantly worse than people with normal weight (*p* = 0.013). The differences between the diet scores between the three compared groups were statistically significant (*p* = 0.027) (Table 7).

## 4. Discussion

Body image perception is an important factor influencing modifiable health risk factors, including eating behaviors. Body weight misperception is understood as the discrepancy between the actual measured weight and the perceived presence of overweight and obesity. This study assessed the diet and selected health parameters in the adult population depending on the perception of body weight.

Over 60% of the surveyed population was overweight or obese as measured by BMI, of which 21.2% were unaware of being overweight and obese. This was particularly common among men (30%). Although the BMI cut-off point for overweight and obesity is 25 kg/m^2^, the subjective point of perception of overweight and obesity based on BMI was 26.4 kg/m^2^ in women and 27.9 kg/m^2^ in men. It also differs in WHR as an indicator for abdominal obesity. The examined women perceived overweight and obesity with a WHR threshold of 0.8 and men with 1.0. The percentage of body fat that characterized participants who considered themselves overweight or obese was 38% for women and 29% for men. Based on this measurement, we diagnose obesity if the body fat content of women exceeds 30% and that of men 25% [26]. The WHR was less associated with one’s perception of overweight or obesity than BMI or percentage of body fat, suggesting also lesser awareness of visceral adiposity and associated with their health risks. Studies have shown that people tend to underestimate their body weight and overestimate their height, especially overweight [26,27]. A nationwide representative survey of adults in the United Kingdom shows that around half of overweight and obese men did not identify as such [28]. Similarly, when analyzing data from a nationally representative survey in the United States of over 16,000 participants, Yaemsiri et al. [29] reported a significant underestimation of weight (48% of men and 23% of women) among participants whose objective weight status was overweight. Compared to overweight people, a lower percentage of obese people (BMI ≥ 30) also find their weight “normal”. One of the reasons overweight participants did not consider their body mass excessive may be that the majority of them did not feel the consequences; shortness of breath and exercise limitations were on a similar level as in non-overweight subjects.

Family history is useful in population-based approaches to disease prevention in high-risk individuals [30]. In our study, participants with a family history of obesity were more often aware of their excess body weight. Despite the existing overweight and obesity, people who perceived their body weight as normal less often recognized this problem in childhood and less often perceive it among siblings. This may also be important for offspring, who might not be perceived as obese by parents and hence not recognize a potential health problem, which will descend to the next generation. In a meta-analysis of over 200,000 people, obese in childhood or adolescence were approximately five times more likely to develop obesity in adulthood compared to those with normal body weight [31].

Carbohydrate metabolism disorders are much more common in people with excess body weight than in the general population [5]. The risk of diabetes in these people increases with the degree of obesity and visceral fat distribution. People unaware of their overweight and obesity had worse body composition parameters than non-overweight individuals, especially higher abdominal circumference and body fat percentage. The assessment of biochemical parameters in our study indicated that people unaware of overweight and obesity had significantly higher total cholesterol levels, LDL and HOMA-IR compared to non-overweight people. This all translates to excessive accumulation of cardiovascular risk factors in individuals with unaware obesity and hence to markedly increased risk of cardiovascular disease. Therefore, we postulate that unawareness of one’s appropriate body mass and composition should be considered as a health problem that may affect the development of cardiovascular disease.

It has been observed that overweight and obese people who misperceive their weight are less likely to control it than those who correctly assess it. A recent study has shown a possible link between perceptions of overweight and obesity, the risk of cardiovascular disease, and body image [32]. It is well known that obesity is an independent risk factor for cardiovascular disease (CVD) and one of the main causes of increased risk of diseases such as dyslipidemia, insulin resistance, hypertension, and atherosclerosis [33,34]. The relationship between obesity and hypertension is well described in adults and across both sexes [35].

Modifying eating behaviors is a key strategy that can prevent obesity [36]. Dissatisfaction with one’s own appearance and body size serves as an incentive to achieve a healthy weight. In our study, people unaware of overweight and obesity positively assessed their own diet, but the evaluation of the actual quality of their diet based on the FFQ turned out to be suboptimal, significantly different from the diet of people with normal body weight, and more similar to obese participants. It seems that people who are satisfied with their own image are less likely to make an effort and change their lifestyle, which is unhealthy and leads to an increased risk of lifestyle-dependent diseases. Proper self-assessment of one’s body image can, in terms of overweight and obesity, influence weight control, lifestyle, and the quality of life [37]. Awareness of the excessive body weight of adults is the basis for the effectiveness of preventive measures. Other studies show that people with obesity who underestimate their weight may not be motivated to change health-related behaviors [38,39].

The limitation of the study is a sample from one region, which may not be fully representative. It would be good to carry out such a study on a more ethnically diverse group. However, we used data collected from a random sample of residents. Moreover, the data on social conditions, medical history, lifestyle, including diet were collected on the basis of the respondents’ answers. This way of collecting data, based on a self-reporting questionnaire, may generate biased information. In the classification of the respondents into groups: people with a normal body weight who considered their body weight to be normal, people with a normal body weight who believed that they were overweight and obese, overweight and obese people who believed that their weight was body is normal, and overweight and obese people who thought they were overweight and obese, we used the BMI index. However, BMI is often used to define weight ranges for the population. We also used the WHR to measure abdominal obesity, which is sometimes insufficient for the proper clinical evaluation of patients. However, the WHR index is a significantly appreciated, easy-to-use meter risk of developing cardiometabolic disorders.

## 5. Conclusions

Our study shows the high prevalence of overweight and obesity and the fact that over 20% of participants did not perceive themselves as such. Underestimation of body weight was associated with higher CV risk factors. The prevalence of underestimation of overweight and obesity in the present study highlights the need for systematic measurements of body mass, waist, and hip circumferences during contacts with healthcare professionals, especially GP visits in a general population. It has to be accompanied by a discussion about the result and its implications for one’s health. This should be followed by self-monitoring of these parameters. People who do not see themselves as obese despite excessive weight have unhealthy eating habits and display higher cardiovascular risk. The problem of underestimation of overweight and obesity was especially frequent among men and people who assessed their nutrition better according to their own opinion. Effective communication about the health consequences of suboptimal weight should be a way to encourage personal weight assessment, evaluation, and possible decisions to support a healthy lifestyle.

## Figures and Tables

**Figure 1 jcm-11-04944-f001:**
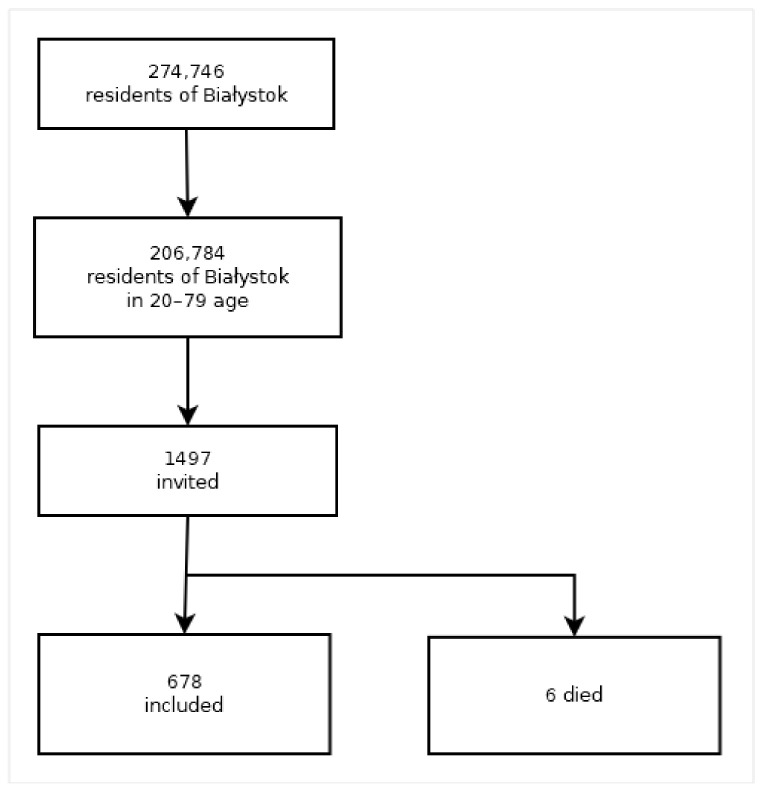
Analyzed cohort.

**Figure 2 jcm-11-04944-f002:**
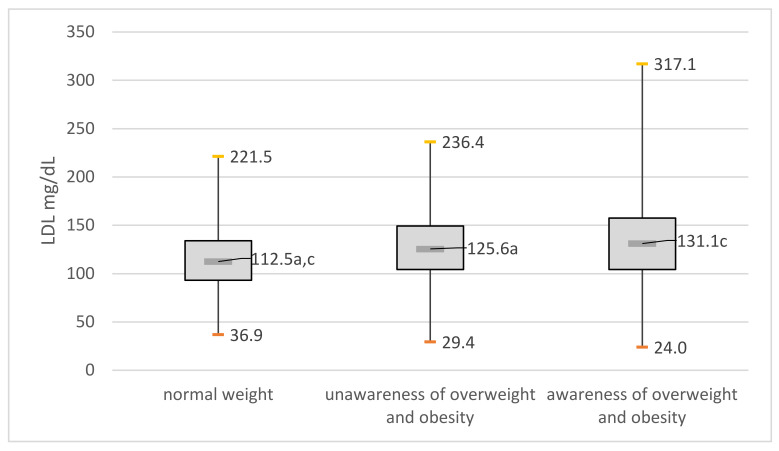
Low-density lipoprotein (LDL) in the compared groups: (a) significant difference between group with normal weight and group unaware of overweight and obesity; (c) significant difference between group normal weight and group aware of their overweight and obesity.

**Figure 3 jcm-11-04944-f003:**
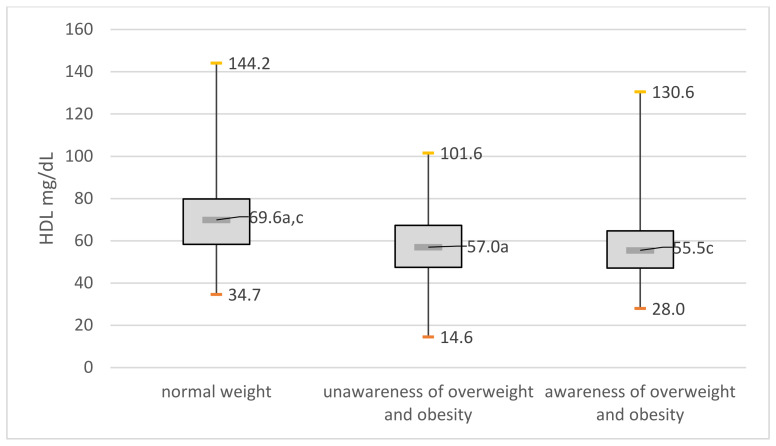
High-density lipoprotein (HDL) in the compared groups: (a) significant difference between group with normal weight and group unaware of overweight and obesity; (c) significant difference between group normal weight and group aware of their overweight and obesity.

**Figure 4 jcm-11-04944-f004:**
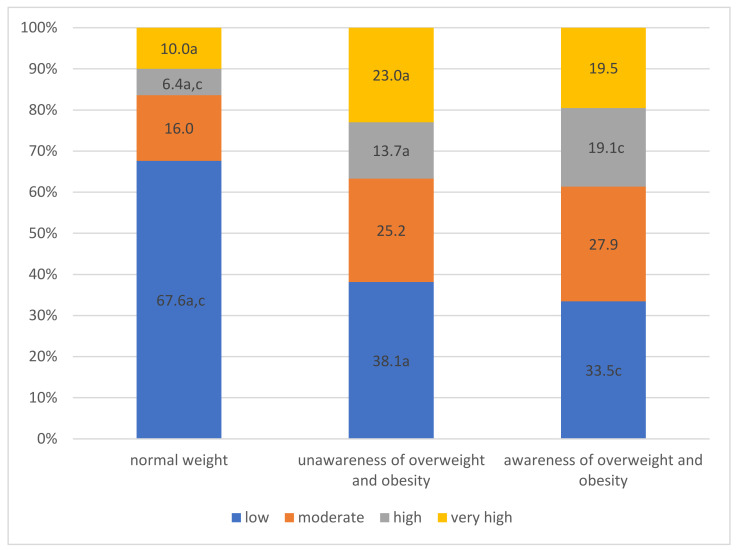
Schematic presentation of cardiovascular risk classes in the study population: (a) significant difference between group with normal weight and group unaware of overweight and obesity; (c) significant difference between group normal weight and group aware of their overweight and obesity.

**Table 1 jcm-11-04944-t001:** Scoring of the consumption of selected products in the study population.

Variables-Food Products	Points		Multiplier 1, 2, −1, −2
	1	2	
Groats	≥once per week <4–6 times per week	≥4–6 times per week	2
Oatmeal, barley flakes, rye flakes, muesli	≥once per week <4–6 times per week	≥4–6 times per week	2
Rye bread	>4–6 times per week <7 times per week	7 times per week	2
Eggs	≥once per week <4–6 times per week	4–6 times per week	1
Milk	≥2–3 times per week <7 times per week	7 times per week	1
Raw vegetables	≥2–3 times per week <7 times per week	7 times per week	2
Boiled vegetables	≥once per week <4–6 times per week	≥4–6 times per week	2
Raw fruits	≥2–3 times per week <7 times per week	7 times per week	2
Legumes	≥once per week <2–3 times per week	2–3 times per week	1
Nuts	≥once per week	2–3 times per week	1
Sea fishes	<2–3 times per week	2–3 times per week	2
Vegetable oils	≥once per week <4–6 times per week	≥4–6 times per week	2
Poultry	2–3 times per week	4–6 times per week	1
Lard, bacon	>once per week <2–3 times per week	≥2–3 times per week	−2
Fatty meat	<once per week >4–6 times per week	≥4–6 times per week	−2
Processed meat	>once per week <4–6 times per week	≥4–6 times per week	−2
Sausages, luncheon meat, canned food	>once per week <4–6 times per week	≥4–6 times per week	−2
Offal	>once per week >4–6 times per week	≥4–6 times per week	−2
Colored carbonated drinks	>once per week <2–3 times per week	≥2–3 times per week	−2
Sweets, cakes, candies, sticks, crackers	>once per week >4–6 times per week	≥4–6 times per week	−2
Fast food	>once per week <4–6 times per week	≥4–6 times per week	−2
Natural spices, herbs	>once per week <4-6 times per week	>=4-6 times per week	2
Bouillon cubes, spice mixtures for dishes	>once per week <4–6 times per week	≥4–6 times per week	−2
Alcohol	>1–2 times per month <2–3 times per week	≥2–3 times per week	−2

**Table 2 jcm-11-04944-t002:** Basic characteristics of the study population.

Variables	Total Population N = 658	Normal Weight (I) N = 256	Unawareness of Overweight and Obesity (II) N = 143	Awareness of Overweight and Obesity (III) N = 259	*p* Value *	Pairwise Comparisons **		
						IvsII	IvsIII	IIvsIII
Sociodemographic characteristic								
Gender (men%)	306 (46.5)	86 (33.6)	92 (64.3)	128 (49.4)	<0.001	sig.		sig.
Age < 40	226 (34.3)	120 (46.9)	42 (29.4)	64 (24.7)		sig.		
Age 40–59	233 (35.4)	87 (34.0)	52 (36.4)	94 (36.3)	<0.001			
Age ≥ 60	199 (30.2)	49 (19.1)	49 (34.3)	101 (39.0)		sig.	sig.	
Education								
Elementary + Vocational based on elementary school	95 (14.5)	27 (10.6)	27 (18.9)	41 (15.8)	0.071			
Secondary + Post-secondary	227 (34.6)	85 (33.5)	44 (30.8)	98 (37.8)				
Higher education	334 (50.9)	142 (55.9)	72 (50.3)	120 (46.4)				
Obesity in family								
Yes	176 (30.9)	57 (24.9)	29 (25.4)	90 (39.6)	0.001		sig.	sig.

The data are shown as N/%, mean ± SD. SD, standard deviation; * Pearson’s χ² test; ** tests for 2 proportions with Bonferroni correction; sig., test result statistically significant after application of Bonferroni correction for multiple comparisons with family-wise error rate of 0.05.

**Table 3 jcm-11-04944-t003:** Anthropometry and body composition of the study population.

Variables	Total Population N = 658	Normal Weight (I) N = 256	Unawareness of Overweight and Obesity (II) N = 143	Awareness of Overweight and Obesity (III) N = 259	*p* Value *	Pairwise Comparisons **		
						IvsII	IvsIII	IIvsIII
Overweight/obesity								
BMI, kg/m^2^	26.8 ± 4.9	22.1 ± 1.9	27.3 ± 4.8	31.1 ± 3.7	<0.001	<0.001	<0.001	<0.001
WHR, cm	0.9 ± 0.1	0.8 ± 0.1	0.9 ± 0.1	0.94 ± 1.2	<0.001	<0.001	<0.001	0.126
BAI, ((HC/height)1.5) − 18	26.8 ± 5.9	23.5 ± 3.9	26.3 ± 4.8	30.5 ± 6.1	<0.001	<0.001	<0.001	<0.001
Body composition								
Average percent fat	32.9 ± 7.5	29.7 ± 6.9	31.1 ± 6.9	37.1 ± 6.1	<0.001	0.393	<0.001	<0.001
Average fat tissue thickness/mm	10.4 ± 1.3	9.1 ± 0.6	10.6 ± 0.7	11.5 ± 0.9	<0.001	<0.001	<0.001	<0.001
Average percent lean mass	63.7 ± 7.7	68.2 ± 6.7	65.2 ± 6.6	58.4 ± 5.7	<0.001	0.001	<0.001	<0.001
Total fat mass/kg	26.1 ± 9.4	17.9 ± 4.7	25.5 ± 5.2	34.5 ± 6.9	<0.001	<0.001	<0.001	<0.001
Total lean mass/kg	49.7 ± 10.9	44.1 ± 8.8	54.2 ± 10.9	52.8 ± 10.5	<0.001	<0.001	<0.001	0.537
Total bone mass/kg	2.7 ± 0.6	2.5 ± 0.5	2.9 ± 0.6	2.8 ± 0.5	<0.001	<0.001	<0.001	0.085
Android fat mass/kg	2.4 ± 1.3	1.3 ± 0.6	2.4 ± 0.7	3.6 ± 1.0	<0.001	<0.001	<0.001	<0.001
Gynoid fat mass/kg	4.0 ± 1.4	3.1 ± 0.9	3.8 ± 0.9	5.1 ± 1.3	<0.001	<0.001	<0.001	<0.001

The data is shown as N/%, mean ± SD. SD, standard deviation; BMI, body mass index; WHR, waist-hip ratio; BAI, body adiposity index; * Kruskal–Wallis test; ** Dwass–Steel–Critchlow–Fligner test.

**Table 4 jcm-11-04944-t004:** Clinical and biochemical characteristics of the study population.

Variable	Total Population N = 658	Normal Weight (I) N = 256	Unawareness of Overweight or Obesity (II) N = 143	Awareness of Overweight and Obesity (III) N = 259	*p* Value *	Pairwise Comparisons *p*-Value **		
						IvsII	IvsIII	IIvsIII
TC/mg/dL	191.3 ± 41.3	183.6 ± 37.7	191.6 ± 40.6	198.5 ± 443.8	0.001	0.102	0.001	0.444
Tg/mg/dL	72.5 ± 29.0	85.3 ± 50.3	119.3 ± 75.4	139.1 ± 79.4	<0.001	<0.001	<0.001	0.003
Fasting glucose/mg/dL	103.6 ± 21.7	96.5 ± 8.9	104.1 ± 18.4	110.4 ± 29.0	<0.001	<0.001	<0.001	0.002
OGTT 120 min-glucose/mg/dL	124.8 ± 39.1	113.7 ± 31.3	124.2 ± 42.8	137.3 ± 40.8	<0.001	0.086	<0.001	0.001
HOMA-IR	3.4 ± 3.1	2.0 ± 0.9	3.2 ± 2.3	4.9 ± 4.1	<0.001	<0.001	<0.001	<0.001
Hba1c/%	5.5 ± 0.7	5.3 ± 0.4	5.6 ± 0.8	5.7 ± 0.8	<0.001	<0.001	<0.001	0.035
BPs/mmHg	125.15 ± 17.77	118.1 ± 16.3	127.7 ± 15.9	130.7 ± 17.9	<0.001	<0.001	<0.001	0.308
BPd/mmHg	81.9 ± 10.4	78.8 ± 9.1	81.5 ± 10.4	85.3 ± 10.7	<0.001	0.026	<0.001	0.002

The data is shown as N (%), mean ± SD. SD, standard deviation; TC, total cholesterol; Tg, triglycerides; OGTT, oral glucose tolerance test; HOMA-IR, homeostatic model assessment of insulin resistance; Hba1c, hemoglobin A1 c; BPs, systolic blood pressure; BPd, diastolic blood pressure; mmHg, millimeters of mercury; * Kruskal–Wallis test; ** Dwass–Steel–Critchlow–Fligner test.

**Table 5 jcm-11-04944-t005:** The perception of overweight and obesity among women and men in ROC analysis.

Variable	AUC	95% C.I.(AUC)	*p* (H_0_: AUC = 0.5)	*		
				Cutoff	Sensitivity	Specificity
Women						
BMI, kg/m^2^	0.9153	(0.887–0.943)	<0.0001	>26.4	81.5%	88.5%
WHR, cm	0.7201	(0.667–0.773)	<0.0001	>0.8	70.5%	65.4%
WC, cm	0.8814	(0.846–0.917)	<0.001	>81.8	80.8%	84.3%
Average percent fat, %	0.8593	(0.822–0.897)	<0.0001	>0.38	80.8%	77.0%
Men						
BMI, kg/m^2^	0.9125	(0.881–0.944)	<0.0001	>27.9	83.8%	86.3%
WHR, cm	0.7679	(0.715–0.821)	<0.0001	>1.0	70.8%	71.4%
WC, cm	0.8778	(0.841–0.915)	<0.001	>90.3	93.1%	66.3%
Average percent fat, %	0.8793	(0.841–0.917)	<0.0001	>0.29	79.2%	83.4%

BMI, body mass index; WHR, waist-hip ratio; WC, waist circumference; centimeters (cm). * criterion—minimizing the distance from the ideal classifier.

**Table 6 jcm-11-04944-t006:** The selected lifestyle elements of the study population.

Variables	Total Population [N/%]	Normal Weight (I) [N/%]	Unawareness of Overweight and Obesity (II) [N/%]	Awareness of Overweight and Obesity (III) [N/%]	*p* Value *	Pairwise Comparisons **		
						IvsII	IvsIII	IIvsIII
Currently smoking:	N = 640	N = 250	N = 136	N = 254				
Yes	126 (19.7)	48 (19.2)	25 (18.4)	53 (20.9)	0.816			
No answer	18 (2.7)	6 (2.3)	7 (4.9)	5 (1.9)				
Self-assessment of diet:	N = 520	N = 205	N=104	N=211				
very good	24 (4.6)	11 (5.4)	9 (8.7)	4 (1.9)	0.001			sig.
good	378 (72.7)	158 (77.1)	80 (76.9)	140 (66.4)				
bad	113 (21.7)	34 (16.6)	15 (14.4)	64 (30.0)				sig.
very bad	5 (1.0)	2 (1.0)	0	3 (1.4)				
no answer	138 (21.0)	51 (19.9)	39 (27.3)	48 (18.5)				

The data is shown as N (%). * Pearson’s χ² test; ** tests for 2 proportions with Bonferroni correction. sig., test result statistically significant after application of Bonferroni correction for multiple comparisons with family-wise error rate of 0.05.

**Table 7 jcm-11-04944-t007:** Average points scored for consumption of recommended products in the study population.

	Q1	Median	Q3	*p* Value
Normal weight	1.0	5.0	10.0	*p* = 0.027
Unawareness of overweight and obesity	−1.0	3.5	9.0	
Awareness overweight and obesity	−1.0	3.0	8.0	
Normal weight vs. unawareness of overweight and obesity *p* = 0.013 Unawareness of overweight and obesity vs. awareness of overweight and obesity *p* = 0.903				

## Data Availability

The data set we generated during analysis is not publicly available due to confidentiality issues but is available from the corresponding author on request.

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
