# Peer review of "How Unawareness of Weight Excess Can Increase Cardiovascular Risk?"

_jcm, 2022, doi:10.3390/jcm11174944_

Round 1

Reviewer 1 Report

This might be an interesting paper, but it is very confusing the way that the results are presented. Results are not matching with methodology. Information here is very useful, but it is not presented clearly. It could be presented by sections and discussed by sections. 

1) Abstract and Introduction 

1a. Line 19 I do not understand how unawareness of obesity is common but it is only 30% in men? So, it should be common in women, right with 70%

1b. Why do you mean by much risk? and Worse? These words are very subjective. 

1c. Introduction must be focused on CVD. 

1e. Why is related the Theory of Visual Normalization?

2. Materials and Methods

2a. It would be useful a table with the methodology of participants followed. 

2b. Line 110- 117 It looks like these would be the main groups along the manuscript, but it is not. Please use the groups along the manuscript

2c. What happened with the group self-assessed as Overweight but presented Normal weight? Where is presented?

2d. Why is using Dwass-Steele-Critchlow-Fligner as a post-hoc test ?

2e. Can we have access at the questionnaire? 

2f. In the questionnaire were included questions about Awareness of Negative Health Consequences?

2e. You were able to evaluate the real diet of the participants?

3. Results. 

3a. It would be useful to specify how the 3 groups in the table were formed.

3b. It is not clear how the P-value is expressed. It is not the correct nomenclature very confusing. It must be changed.

3c. Line 169-171 just rewording the table 2. 

3d. Where the family history is presented? What table or figure? I could not find it. 

3e. Where are the ROC Curves figures?

4. Discussion

4a. It is hard to follow the discussion with results so confusing.

4b Where is the information about family history? 

5 Conclusions

5a. Conclusions are not related to the manuscript

6 References

6a. References are not well cited. For example No. 22

Author Response

Dear Reviewer,

Please, see the atachment.

Kind regards,

Magdalena Zalewska

Reviewer 2 Report

The authors presented interesting results of the study of analysis of people affected by the misperception of excess body weight and their eating behaviors simultaneously. I have some comments:

1)      The title is very expressive, but still does not adequately reflect the main essence of the study.

2)      Specify whether all quantitative variables had a normal distribution? This may affect the correctness of the choice of descriptive statistics, methods of statistical analysis, as well as the results.

3)      The authors mentioned visceral obesity twice in the discussion. That's a good thing. However, in general, this aspect is missed in the study design, despite the high importance of visceral obesity for cardiovascular risk. Authors used waist circumference and WHR in the study. Indeed, these parameters have been the most commonly used anthropometric parameters to identify and quantify intra-abdominal fat deposition. There are characterized by simplicity, low cost, and acceptable accuracy. However, an intra- and inter-examiner variability >3% limits its usefulness in monitoring slight variations in abdominal adiposity; also, these indices seems to quantify subcutaneous fat better than visceral fat. CT or US methods are more preferable for visceral obesity assessment. It is necessary to discuss the relevant limitations of the study.

Author Response

(The authors gave the same response as above.)

Round 2

Reviewer 1 Report

Dear authors,

Thank you for your responses. In my opinion, manuscript has improved after reviewing all the comments. I still think that this manuscript must be dived in 2 different manuscripts or divided the sections in the results and discussions to be more easy to read.

Still some points are unclear

Line 23 what do you mean by much higher any percent? Any fold? It is necessary to be specific

Why is using Dwass-Steele-Critchlow-Fligner as a post-hoc test ? Any definition of this test? It must be based on scientific evidence.

Regarding my question 3b and 3c from revision 1, where are those changes? Please specify

Regarding my question 3d and 3e from revision 1, the table about obesity in Family history. It is going to be included in the supplemental material. Normal readers are going to have access to this information?

Regarding conclusions, still are not related to the manuscript findings.

Some references still are not cited corrected.  I could not find reference 1 on the web. Reference 36, I could not find it.

Reviewer 2 Report

I approve revised version of paper.

Author Response

Dear Reviewer,

Thank you for approve revised version of our paper.

Kind regard

Magdalena Zalewska